# Niche-processes induced differences in plant growth, carbon balance, stress resistance, and regeneration affect community assembly over succession

**Zhaoyuan Tan**[1,2], **Hui Zhang**[1,2]*

**1** College of Forestry, Wuzhishan National Long Term Forest Ecosystem Monitoring Research Station, Hainan University, Haikou, P. R. China, **2** Key Laboratory of Genetics and Germplasm Innovation of Tropical Special Forest Trees and Ornamental Plants (Hainan University), Ministry of Education, College of Forestry, Hainan University, Haikou, P. R. China

* zhanghuitianxia@163.com

**Data Availability Statement:** All data underlying the study are available within the paper and its Supporting Information files.

## Abstract

The relationship between plant traits and species relative abundance along environmental gradients can provide important insights on the determinants of community structure. Here we bring extensive data on six key traits (specific leaf area (SLA), seed mass, seed germination rate, height, leaf proline content and photosynthesis rate) to test trait-abundance relationships in a successional chronosequence of subalpine meadow plant communities. Our results show that in late-successional meadows, abundant species had higher values for seed mass, seed germination rate, and SLA, but had lower values for height, photosynthesis rate, and leaf proline content than rarer species. The opposite patterns of trait-abundance relationships were observed for early-successional meadows. Observations of strong trait convergence and divergence in these successional communities lend greater support for niche processes compared to neutral community assembly. We conclude that species' niches that determine plant growth (plant height and photosynthesis rate), carbon balance (SLA, photosynthesis rate), regeneration (seed mass and seed germination rate), and abiotic stress resistance (leaf proline content) under different environmental conditions have strong influence on species relative abundance in these sub-alpine meadow communities during succession.

## Introduction

What factors determine species abundance or why some species are common and others are rare during ecological succession are among the oldest questions in community ecology [1–4]. The mechanisms that determine spcies abundance are reflected in differentiating the relative importance of neutral versus niche-assembly processes in determining species abundance [5]. Quantifying the role of functional traits on species abundance is central to the evaluation of the neutral and niche processes that shape species abundance [6, 7]. Niche theories invoke species-level differences in functional traits that represent evolutionary adaptations to abiotic and biotic environments [8]. Species abundance is thus determined by a series of unavoidable

**Funding:** This work was funded by the start-up fund from Hainan University (KYQD (ZR) 1876). Z. Y.T. and H.Z. designed research; Z.Y.T. and H.Z. performed research; H.Z. analysed data; Z.Y.T. and H.Z. wrote the paper.

**Competing interests:** The authors have declared that no competing interests exist.

trade-offs that reflect abiotic and biological constraints on life history evolution [5]. Neutral theories assume that trophically similar species have equivalent odds of becoming abundant, irrespective of their functional attributes [6]. Trait differences among species may thus persist in neutral communities as long as they do not lead to differences in per capita demographic rates and fitness [9]. Therefore relating trait-abundance patterns within a community may not reveal the contributions of neutral or niche processes to community structure [5, 10]. However, when communities change during succession, the shifts in trait-abundance relationships during community development over time (or space) may provide a better opportunity to evaluate the contributions of niche and neutral mechanisms. This is because niche and neutral processes predict very specific directional changes during succession, which can be tested.

At the global scale, plant distributions or species presence and absence are undoubtedly determined by plant functional attributes that influence physiological tolerance or resource requirements [11]. Tests of inter-specific differences in traits and their trade-offs among the suite of co-occurring species along different environment gradients [12, 13] show that trait-correlations of species presence or absence are ubiquitous across different biomes [14, 5]. However, the observation that traits affect the presence/absence of species does not necessarily imply that traits can determine which species are abundant and rare, or species relative abundance itself [5]. Even though broader investigations on the influence of ecological and physiological factors on species abundance across environmental gradients have been undertaken [12, 13, 5, 15], the relationship between traits and species abundance is not easily derived. Moreover, there are two following questions remains to be explored for current trait-abundance relationship studies for increasing the predictable power of traits for community assembly processes.

First, most functional-trait based studies have tended to select traits that are easy to measure for multiple species and sites. Traits such as specific leaf area (SLA), leaf dry matter content (LDMC), and plant height are relatively easy to measure but may not directly determine plant function and fitness in a given environment. Here, eco-physiological traits such as plant photosynthetic capacity and abiotic stress tolerance, which are relatively hard to measure, can provide direct measurements of life history strategies, but are usually not studied [16, 17]. Actually, the easily-measured traits are generally only proxies for key physiological traits [18], therefore a wide range of species traits (not just easily measured traits) should be considered for trait-based tests of community assembly mechanisms [19].

Second, quantifying the importance of neutral processes on community assembly using trait-abundance relationships is difficult. Although neutral processes do not predict any trait–abundance connection [6], any observed correlation between traits and abundance does not preclude the dominant influence of neutral processes on abundance either [5]. That is because, some traits (e.g., seed mass and seed germination rate) can also highly related to neutral processes (dispersal and recruitment limitation) [20, 21]. Functional trait diversity (FD) patterns are however more strongly determined by the nature of niche and neutral processes. So, if traits do not determine species presence/absence or abundance at a site, the diversity in trait values at the site should simply reflect that of a random sample from the larger species pool. On the other hand strong environmental filtering would select for a narrower range of trait values resulting in trait convergence. If resource competition were dominant, dissimilar species would be selected at a site causing trait divergence. Appropriate expectations can then be derived for FD patterns under purely neutral assembly (FDrandom), against which the observed FD patterns (FDobserved) can be compared. Thus testing FD patterns during succession can further facilitate to find out whether neutral or niche processes determine the variations in functional traits thereby to differentiate the relative importance of niche and neutral processes in species abundance.

Our study site is located in the sub-alpine meadow community in the Qinghai-Tibetan Plateau. It is featured by a chronosequence of meadows that range from "natural" (undisturbed for at least 40-year) to those that have been protected from agricultural exploitation for 4, 6, 10, and 13 years, respectively. We also note that the environment in the plateau is characterized by intense and prolonged UV radiation, extremes of temperature, short growing season, and low soil fertility, which may influence important physiological traits (e.g. photosynthesis rate and abiotic stress resistance) [22, 23]. Such observations have been made on species that occur in harsh environments [24], therefore we expect strong relationships between species relative abundance and abiotic stress tolerance. Here we attempt to utilize assembling data on six easy-to-measure morphological (specific leaf area (SLA), height and seed mass) and hard-to-measure physiological traits (seed germination rate, leaf proline content and photosynthesis rate) to test trait-abundance relationships in a successional chronosequence of subalpine meadow plant communities. By doing this, we can reveal which life history strategies that are reflected by these 6 six traits can determine species abundance during succession. Since seed mass and seed germination rate are highly associated with neutral processes [20, 21],we also quantify whether FD is random or not along succession to check whether neutral processes can lead to variations in functional traits.

## Materials and methods

### Sampled sites

Our chronosequences are located in the meadows near Hezuo city, China (34˚55′N, 102˚53′E), which is at the eastern edge of the Qinghai Tibetan plateau. The mean elevation of our study sites is 3050 m above sea level. Climatically, the site is cold and dry, with mean annual temperature of 2.4˚C and mean annual precipitation of 530 mm (distributed mostly in July and August). The dominant species in the natural meadow include *Elymus nutans*, *Kobresia pygmaea* (C. B. Clarke), and *Thermopsis lanceolata* (R. Br) [25].

### Field sampling

Sampling was conducted in August (the peak growing season) of 2013 in meadows with five different successional ages (4-, 6-, 10-, 13-year, and undisturbed for at least 40-year). Within a large landscape near Hezuo city, we identified two spatially distinct study areas with the same successional chronosequences (named chronosequence 1 and chronosequence 2) for which farming histories might be reliably obtained by interviewing local farmers. These two areas (chronosequence 1 and chronosequence 2) were ~10 km apart and had comparable topographic characteristics (e.g., orientation and slope). We therefore had two independent replicates for each successional age, yielding a total of 10 sites (successional meadows). At each site we randomly selected an area of 120 ×120 m$^2$ and subsequently arranged 30 quadrats (0.5 × 0.5 m$^2$) regularly in six parallel transects, with 20 m intervals between adjacent quadrats. We enumerated total aboveground ramets of each species to quantify abundance in each of the 30 quadrats for each successional meadow. Specifically all field sites access were approval by the research station of alpine meadow and wetland ecosystem of Lanzhou University.

### Collection of functional traits

We quantified the carbon economy of leaves by measuring SLA (leaf fresh area per dry mass). We quantified light capture strategy via maximum plant height and photosynthesis rate. We measured traits relating to abiotic stress resistance (leaf proline content), and regeneration (seed mass and germination rate). Importantly we measured the traits of the same species at

each successional age separately if they occurred in multiple meadows, ensuring that intra-specific variation was properly incorporated in our analyses. The detailed field methods are given in the S1 Table.

## Statistical methods

**Trait-abundance relationships over succession.** Our primary hypothesis is whether the relationship between traits and abundance shifts with successional age. The appropriate test for this is ANCOVA, with abundance as the dependent variable, successional age as a fixed, discrete factor and the trait as a continuous covariate. Here we use the ANCOVA model to examine systematic changes in trait-abundance relationships with successional age. Successional age was a grouping factor that consisted of five levels. For each trait, we used the following model:

$$Abundance_{ijkl} = Trait_i + Age_j + Trait_i \times Age_j \quad (1)$$

where Abundance$_{ijk}$ is $i$th species density (individuals per plot); Trait$_i$ is $i$th species trait value; Age$_j$ ($j$ = 4, 6, 10, 13 and undisturbed) is age class.

A significant successional age × covariate interaction would indicate that the slope of the trait–species relative abundance relationship differs among successional ages. In ANCOVA residuals should follow normal distribution, not the covariates, Thus, before performing ANCOVA analysis, we log-transformed species abundances and functional trait values to normalize the data. We expect that dominant and rare species exhibit different values for our measured traits with increase in successional age.

**Comparing the relative contributions of niche and neutral processes to community assembly during succession.** We tested FD patterns for all measured functional traits for each successional meadow community thereby revealing the relative contribution of neutral and niche processes to community assembly during succession. Several methods have been described and discussed on how to calculate the functional trait diversity (FD) [26]. Among these, we chose Rao's quadratic entropy (RaoQ), which is an efficient functional diversity index, because it is an intuitive generalization of the Simpson's index of diversity, and is easily interpretable [26]. To test whether any observed FD pattern is a random distribution, or niche-based processes induced trait convergence and trait divergence at each successional meadow community, we first simulated null communities in which species and trait values were randomly distributed. Randomization procedures were applied to calculate "null" distributions for both species composition (i.e., on the species × quadrat matrix) and FD [27]. Reshuffling the species × quadrat matrices was done with three constraints, i.e., keeping: i) the same number of species (species richness) per plot in the simulated and observed data; ii) the same number of total species occurrences per region (i.e., number of plots where the species occur in a region); and iii) the total abundance of species in a region constant (i.e., the sum of the number of quadrats occupied in all plots). We implemented this using the function "randomizeMatrix" in the "picante" package in R [28]. We then compared the observed FD to the FD simulated in 1000 randomly assembled communities. Specifically, we computed the standard effect size index (SES) following Gotelli & McCabe [29]:

$$SES = \frac{FDobs - FDrandom}{FDsd} \quad (2)$$

where FDobserved and FDrandom represents observed FD and mean FD values of the simulated null community, respectively. FDsd represents the standard deviation of FD values generated from the 1000 simulations. It should be noted that, when FDrandom cannot meet the

symmetry distribution, FDobs an FDrandom should be log-transformed [30]. Thus here we firstly log-transformed FDobs an FDrandom and then we calculated SES. We used Wilcoxon signed-rank tests to examine whether SES is significantly more than, less than or equal to zero, which indicates the prevalence of significant trait divergence, trait convergence, and random distribution (neither trait convergence nor divergence), respectively. Our confidence in the findings is enhanced given that we found consistent patterns in two independent sites for each successional phase. As a result, we tested trait-abundance relationships and FD patterns over succession for chronosequnce 1 and 2 individually to see whether consistent patterns can be found for both chronosequence 1 and 2.

## Results

The ANCOVA model results show that the correlations between species relative abundance and six key eco-physiological traits (plant height, photosynthesis rate, leaf proline content, SLA, seed mass, and seed germination rate) shift significantly with increase in successional age at both sites (p<0.05, Tables 1 and 2). However, only SLA, and leaf proline content vary significantly with increase in successional age at both sites (p<0.001, Fig 1).

We then examined the patterns of these trait-abundance relationships. We found that plant height, photosynthesis rate and leaf proline content were positively associated with species abundances at early phases (4- and 6-year meadow), but were negatively related species abundances in late states (10-, 13-year and undisturbed meadow) at both chronosequences (p<0.05, Fig 2). In contrast, SLA, seed mass and seed germination rate were negatively associated with abundance in early, but were positively related species abundances in late succession at both chronosequences (p<0.05, Fig 3).

Finally, we tested functional trait diversity patterns during succession and found significant trait convergence, i.e., the SESs were significantly less than zero, for the 4-year and late successional (10-year, 13-year, and undisturbed) site replicates (Fig 4), and trait divergence, i.e., the SESs were significantly greater than zero for the 6-year successional replicate sites (Fig 4).

## Discussion

### The observed shifts in plant height, photosynthesis rate, SLA, and seed mass along the successional gradient

We found evidence for significant changes in plant height, photosynthesis rate, SLA, and seed mass exhibited by the dominant species (high relative abundance) along the successional gradient. These relationships between eco-physiological traits (SLA, plant height, photosynthesis rate and seed mass) and species abundance support previous findings of directional changes in traits along successional gradients [31, 32, 23].

At the early stages of succession, species with small seeds, high photosynthetic rates, rapid growth rate and short life-span are likely to invade and dominate communities [33, 34]. As succession proceeds, the amount of light reaching the ground, available soil nutrients, and decomposition rates of plant material all decline [32]. Abundant species in late successional communities are predicted to be superior competitors for those limiting resources [31]. Based on the prevalent trade-off between seed mass and the number of seeds produced per plant, species producing a great number of small seeds are more likely to colonize vacant sites [31, 35, 23]. However, as succession progresses, larger-seeded species slowly establish and become abundant, presumably due to greater seed investments resulting in higher survival and competitive ability [21, 5] and greater stress tolerance [36].

**Table 1. The relationships between species abundance and functional traits (SLA, seed mass, seed germination rate, plant height, photosynthetic rate and leaf proline content) with successional age in chronosequence 1, using ANCOVA model (formula: abundance ~ Trait + Age + Trait × Age).**

| Source | F/P | Age* | Trait | Trait × Age |
|---|---|---|---|---|
| | | (4) | (1) | (4) |
| SLA | F | **4.50** | **5.09** | **13.51** |
| | P | **<0.05** | **<0.001** | **<0.001** |
| Seed mass | F | 0.35 | 0.80 | **5.71** |
| | P | >0.05 | >0.05 | **<0.001** |
| Seed germination rate | F | 3.16 | 1.78 | **4.481** |
| | P | >0.05 | >0.05 | **<0.001** |
| Plant height | F | 0.01 | 0.34 | **3.92** |
| | P | >0.05 | >0.05 | **<0.05** |
| Photosynthetic rate | F | **12.50** | 0.55 | **2.98** |
| | P | **<0.001** | >0.05 | **<0.001** |
| Leaf proline content | F | 0.09 | 0.96 | **3.52** |
| | P | >0.05 | >0.05 | **<0.001** |

The F statistics and corresponding P values are shown for the six functional traits. Boldface type indicates significant differences at P < 0.05.

## The role of plant abiotic stress resistance in determining species composition along the successional gradient

It is known that environmental factors such as drought, nutrient imbalance, temperature stress, and UV radiation exposure will select for plant abiotic stress resistance that enhances plant growth and survival in these conditions [37]. Due to intense and prolonged UV radiation exposure, extremes of temperature, and poor soil fertility in the Qinghai Tibetan plateau [24], abiotic stress tolerance may be a major factor that affects species composition under the harsh environmental conditions. As one of the major organic osmolytes, proline usually accumulates in a variety of plant species in response to environmental stresses, thus proline content may be a valuable indicator describing the response and/or adaptation of plants to stresses [38].

**Table 2. The relationships between species abundance and functional traits (SLA, seed mass, seed germination rate, plant height, photosynthetic rate and leaf proline content) with successional age in chronosequence 2, using ANCOVA model (formula: abundance ~ Trait + Age + Trait × Age).**

| Source | F/P | Age* | Trait | Trait × Age |
|---|---|---|---|---|
| | | -4.00 | -1.00 | -4.00 |
| SLA | F | 0.03 | 1.34 | **8.94** |
| | P | >0.05 | >0.05 | **<0.001** |
| Seed mass | F | 0.07 | **2.87** | **3.74** |
| | P | >0.05 | **<0.05** | **<0.001** |
| Seed germination rate | F | **11.72** | **5.82** | **7.28** |
| | P | **<0.001** | **<0.001** | **<0.001** |
| Plant height | F | 0.37 | 2.21 | **4.04** |
| | P | >0.05 | >0.05 | **<0.001** |
| Photosynthetic rate | F | 3.47 | **2.97** | **5.43** |
| | P | >0.05 | **<0.05** | **<0.001** |
| Leaf proline content | F | 0.48 | **6.15** | **8.86** |
| | P | >0.05 | **<0.001** | **<0.001** |

The F statistics and corresponding P values are shown for the six functional traits. Boldface type indicates significant differences at P < 0.05.

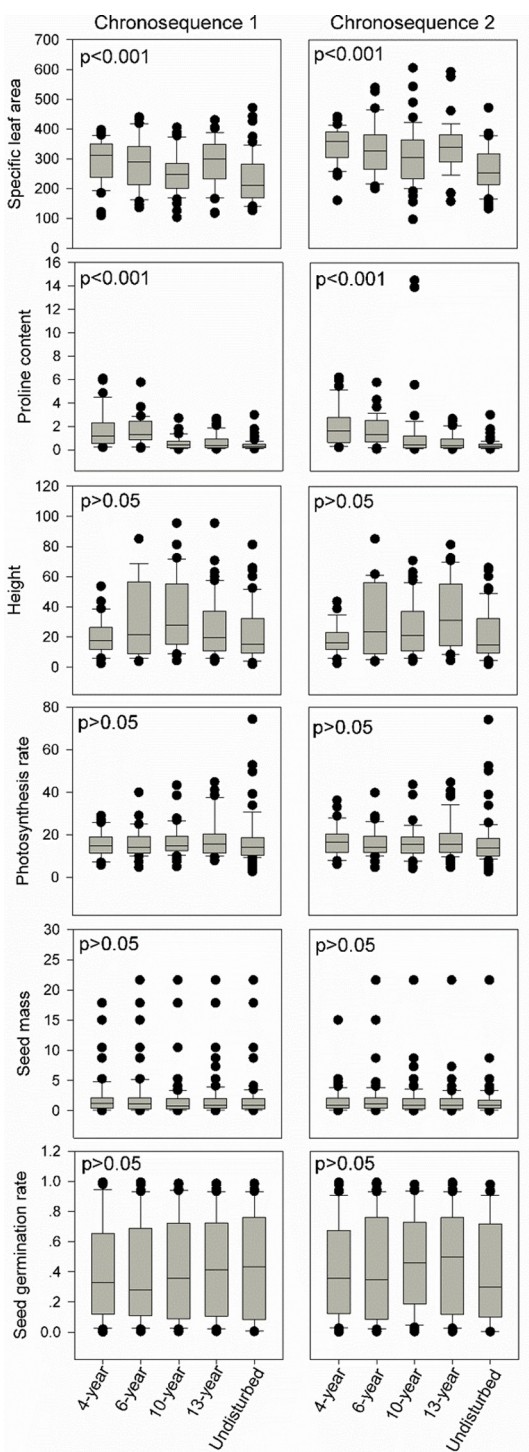

**Fig 1. Boxplot of the values for the six functional traits (specific leaf area, leaf proline content, height, photosynthesis rate, seed mass and seed germination rate) in chronosequence 1 and 2 respectively.**

Indeed, we found significant leaf proline content-species abundance relationships over succession, i.e., leaf proline content was highly positive correlated to species abundance in early successional states (abandoned 4- and 6-year meadow), but negatively associated with species

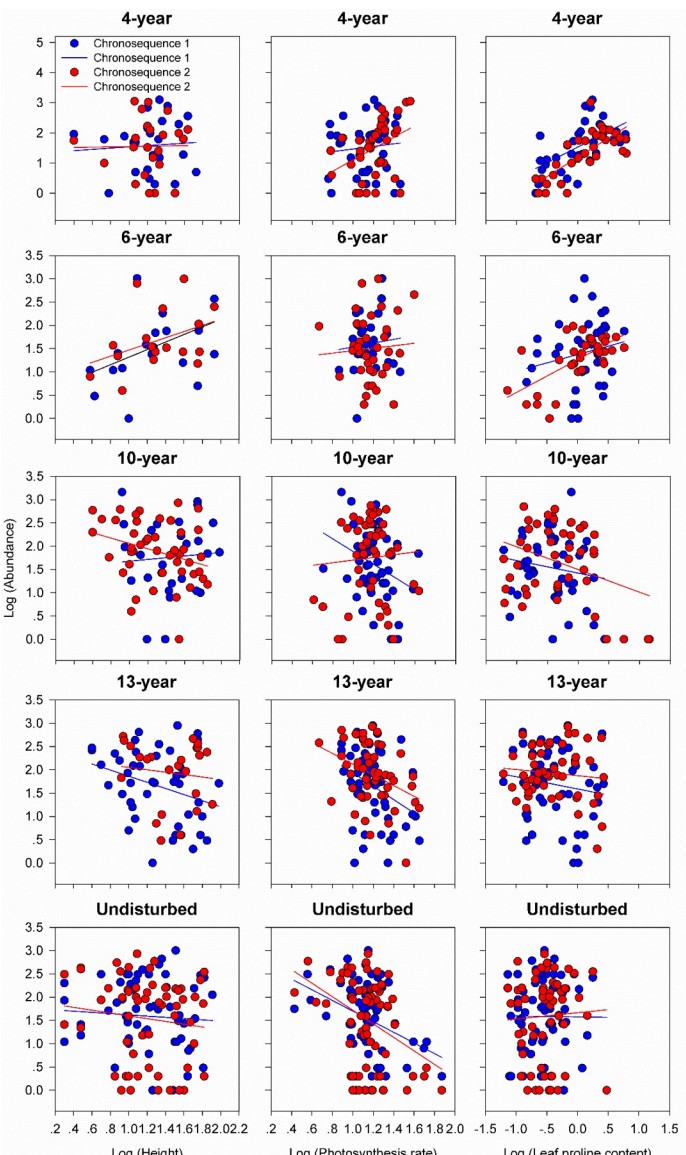

**Fig 2. The relationship between the abundance of each species in each meadow and its functional traits (plant height, photosynthesis rate and leaf proline content) with successional age in chronosequence 1 and 2 respectively.** Each point represents the mean value of abundance and funtinal traits for a single species. Fitted lines are generated from ANCOVA model (abundance~traits + Age + traits × Age) changing from positive values in early-successional meadows (4- and 6-year) to negative values in late successional meadows (10-, 13-year and undisturbed).

abundance in the late successional meadows. This appeared to indicate that species exhibiting abiotic stress resistance or adaptation to stress tend to dominate at earlier successional stages.

## The relationship between seed mass and germination rate along the successional gradient

In natural environments, suitable conditions for germination and seedling establishment may occur unpredictably both in time [39] and space [40]. Compared with the late successional stage, plants face the highest risks of mortality in the early stage of succession, because of preferential investment in growth. Together with the relatively high availability of suitable sites for

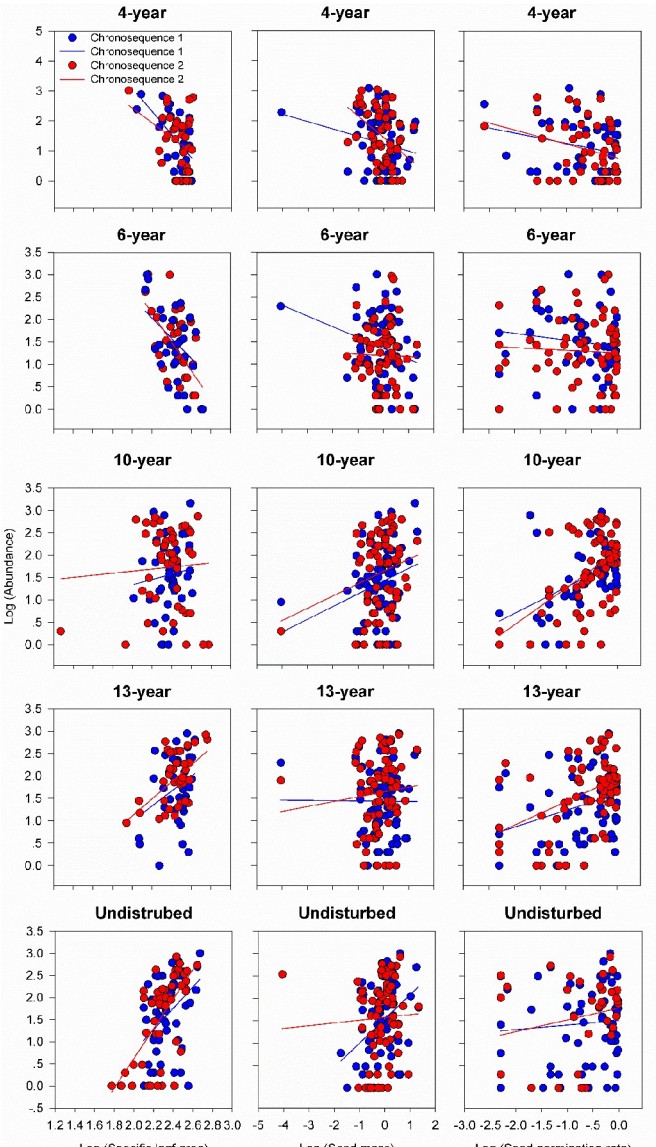

**Fig 3. The relationship between the abundance of each species in each meadow and its functional traits (specific leaf area, seed mass and seed germination rate) with successional age in chronosequence 1 and 2 respectively.** Each point represents the mean value of abundance and funtinal traits for a single species. Fitted lines are generated from ANCOVA model (abundance~traits + Age + traits × Age) changing from negative values in early-successional meadows (4- and 6-year) to positive values in late successional meadows (10-, 13-year and undisturbed).

establishment, reproductive allocation may favor the production of large numbers of small seeds compared to fewer larger seeds [41]. Thus, seed mass may represent one of the most important traits influencing the early phases of the plant life cycle, including germination [42], emergence [43], growth and survival of seedlings [44]. Theoretical models predict that large-seeded species should germinate more rapidly than small seeded species, in that large seeds are more likely to have higher post-dispersal seed predation than small seeds [45, 46]. In the present study, we found that in the early stages of succession, dominant species tended to have small seeds and low germination rates, while species that possess greater seed mass and high germination rate were common in the late successional meadows. These observations are

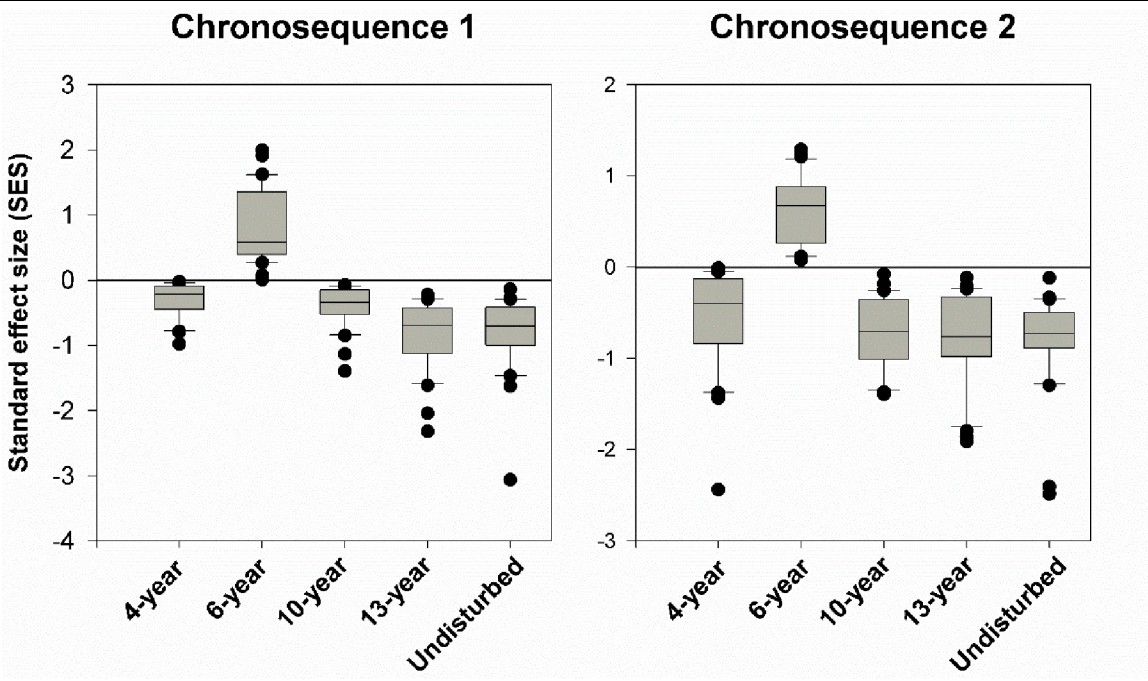

**Fig 4. The distributions of standard effect size (SES) of the differences between observed FD and FD generated from 1000 random communites during succession.** SES that is significantly greater than, smaller than, or approaching zero indicates significant trait divergence, trait convergence, or random distribution (neither trait convergence nor trait divergence), respectively. Box plots show the median (line within the box), 25th and 75th percentiles (the boundaries of the box) and 90th and 10th percentiles (error bars) of SES at each successional age in chronosequence 1 and 2 respectively. *** indicates P<0.001 based on Wilcoxon signed-rank tests.

consistent with the results derived from theoretical models that large seeds should germinate faster than small seeds [45, 46].

### The effects of niche processes on community assembly along the successional gradients

The significant shifts in trait-abundance relationships over succession clearly indicated that selection for plant life history attributes is linked to niche-based processes (e.g., abiotic filtering and resource competition) over succession. However, since seed mass and seed germination rate are also highly related to neutral processes [20, 21], thus these observed connections between traits and abundance do not preclude the contributions of neutral processes such as drift and demographic stochasticity to community assembly [5]. However, our FD patterns preclude this possibility. That is because SES were significantly greater or less than zero over succession, suggesting that niche processes but not neutral processes prominently determined the variations in functional traits. Thus niche processes gave rise to the variations in our measured six functional traits, which in turn determined species abundance over succession.

### Conclusion

Our results constituted a significant advance in connecting several traits, including both easily measured morphological traits and difficult-to-measure physiological and reproductive traits to community structure. We found a dominant role for niche processes but not neutral processes in determining the variations in our measured six funtional traits which in turn determined species abundance during succession. Since these six traits were related to growth and competitive ability, stress tolerance, and dispersal, which were important aspects of plant life

histories, species abundance during succession was mainly driven by niche-processes that selected for plant attributes such as growth, carbon balance, stress resistance, and regeneration.

## Supporting information

**S1 Table. The long-transformed values of abundance (abundance) and mean value of traits (Value) for each species found in each successional age in chronosequence 1 and 2, including specific leaf area (SLA; g cm$^{-2}$), seed mass (SM; g), seed germination rate (SG; %), plant height (H, cm), photosynthesis rate (A; μmol m$^{-2}$ s$^{-1}$) and leaf proline content (Pro; mg/kg).** The values for fitted ANCOVA lines (FAL) are also shown.
(DOC)

## Acknowledgments

We would also like to thank Christine Verhille at the University of British Columbia for her assistance with English language and grammatical editing of the manuscript.

## Author Contributions

**Data curation:** Zhaoyuan Tan, Hui Zhang.

**Formal analysis:** Zhaoyuan Tan, Hui Zhang.

**Funding acquisition:** Hui Zhang.

**Investigation:** Hui Zhang.

**Methodology:** Hui Zhang.

**Software:** Hui Zhang.

**Supervision:** Hui Zhang.

**Validation:** Zhaoyuan Tan, Hui Zhang.

**Visualization:** Zhaoyuan Tan, Hui Zhang.

**Writing – original draft:** Zhaoyuan Tan, Hui Zhang.

**Writing – review & editing:** Zhaoyuan Tan, Hui Zhang.

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
