## [Decision Letter · Decision Letter 0]

2 Jan 2020

PONE-D-19-21921

Niche-processes induced differences in plant growth, carbon balance, stress resistance, and regeneration affect community assembly over succession

PLOS ONE

Dear Mr Zhang,

Thank you for submitting your manuscript to PLOS ONE. After careful consideration, we feel that it has merit but does not fully meet PLOS ONE’s publication criteria as it currently stands. Therefore, we invite you to submit a revised version of the manuscript that addresses the points raised during the review process.

The review process took longer than expected because a third review was sought to resolve the conflict between the first two reviewers.  As the third review was positive, the decision was made not to reject the manuscript for publication, rather to return it for revision. Please pay careful attention to the suggestions and criticism from all three reviews.

We would appreciate receiving your revised manuscript by Feb 16 2020 11:59PM. To enhance the reproducibility of your results, we recommend that if applicable you deposit your laboratory protocols in protocols.io, where a protocol can be assigned its own identifier (DOI) such that it can be cited independently in the future. For instructions see: http://journals.plos.org/plosone/s/submission-guidelines#loc-laboratory-protocols

We look forward to receiving your revised manuscript.

Kind regards,

Craig Eliot Coleman, PhD

Academic Editor

PLOS ONE

Journal Requirements:

Reviewers' comments:

Reviewer's Responses to Questions

**Comments to the Author**

1. Is the manuscript technically sound, and do the data support the conclusions?

Reviewer #1: No

Reviewer #2: No

Reviewer #3: Yes

2. Has the statistical analysis been performed appropriately and rigorously? 

Reviewer #1: No

Reviewer #2: No

Reviewer #3: Yes

3. Have the authors made all data underlying the findings in their manuscript fully available?

Reviewer #1: No

Reviewer #2: Yes

Reviewer #3: Yes

4. Is the manuscript presented in an intelligible fashion and written in standard English?

Reviewer #1: No

Reviewer #2: Yes

Reviewer #3: Yes

5. Review Comments to the Author

Reviewer #1: Understanding rules of community assembly is a central goal of community ecology, and trait-based models are useful tools to reach this goal. Therefore, the topic of this paper is interesting. Its further strength that the authors measured hard traits. Unfortunately, I have found so many problems in the manuscript that overwhelm the above mentioned advantages.

The logic of the Introduction and aims of the paper should be re-think. The coexistence of totally neutral species is stable, but not robust, because nearly neutral species cannot coexist. For more details on this topic see e.g. (Meszéna et al. 2006; Barabás et al. 2018). Thus, studying role of neutrality is meaningless, however some traits may prove to be neutral. Possibly, the authors mixed neutrality and stochasticity (it often occurs in the literature). Of course, exploring the relative role of stochastic and deterministic processes could be a valid aim. However, it would need different methodology, e.g. CATS developed by Shipley (Shipley et al. 2012; Shipley 2014). The applied methods allow quantifying importance of habitat filtering and limiting similarity, and its change during succession. Either the aims should be reformulated or different methods should be chosen.

I am missing the mentioning studies of community assembly along successional series from the Introduction. Moreover, in some references I think that the cited paper did not support well the statement (sometimes it did not say it at all, other cases it is not the main message of that paper and better citation could be found). For example, Ref3 in line 41; Ref8 in line 54; in line 55 paper that study changing trade-off along gradients also should be cited; Ref9 in line 61; Ref13 in line 74; Ref14 in line 78.

In the Methods, it remains unclear for me exactly how many plots were sampled and how they were arranged. It only mentioned later that two chronosequences were sampled.

I think the assumptions of ANCOVA (independence and normality of errors) were not satisfied. I suggest consulting with (Warton et al. 2015) and references therein. If abundance were measured by number of ramets, Poisson or negative binomial distribution would be suitable. Note, that in ANCOVA residuals should follow normal distribution, not the covariates.

To support your selection of randomization constrains please see (Götzenberger et al. 2016), and also see my paper (Botta-Dukát 2018) on possible problems of Gotelli & Grave’s SES.

Because correcting the above mentioned problems will lead to considerable changes in Results and Discussion, I stopped reviewing the manuscript at the end of Methods section.

Cited references:

Barabás G, D’Andrea R, Stump SM (2018) Chesson’s coexistence theory. Ecol Monogr 88:277–303. doi: 10.1002/ecm.1302

Botta-Dukát Z (2018) Cautionary note on calculating standardized effect size (SES) in randomization test. Community Ecol 19:77–83. doi: 10.1556/168.2018.19.1.8

Götzenberger L, Botta-Dukát Z, Lepš J, et al (2016) Which randomizations detect convergence and divergence in trait-based community assembly? A test of commonly used null models. J Veg Sci 27:1275–1287. doi: 10.1111/jvs.12452

Meszéna G, Gyllenberg M, Pásztor L, Metz JAJ (2006) Competitive exclusion and limiting similarity: A unified theory. Theor Popul Biol 69:68–87. doi: 10.1016/j.tpb.2005.07.001

Shipley B (2014) Measuring and interpreting trait-based selection versus meta-community effects during local community assembly. J Veg Sci 25:55–65

Shipley B, Paine CET, Baraloto C (2012) Quantifying the importance of local niche-based and stochastic processes to tropical tree community assembly. Ecology 93:760–769. doi: 10.1890/11-0944.1

Warton DI, Shipley B, Hastie T (2015) CATS regression – a model‐based approach to studying trait‐based community assembly. Methods Ecol Evol 6:389–398. doi: 10.1111/2041-210X.12280

Reviewer #2: The authors examined the trait-abundance relationships and trait diversity patterns among successional stages in two meadow communities. The dataset of both traits and communities gives good opportunities to explore the ecological processes. However, there are some very important issues must be clarified.

First, for the analyses of trait-abundance relationship, the authors used the model: trait = abundance+age +abundance:age. I am very confused why abundance can predict the functional traits. Also, most parts of Introduction describe how traits can influence species abundance (I agree and this is reasonable). If the true model the authors wanted to use is abundance = trait + age + trait:age, then all relevant results needed to be revised. Also, the results are not consistent with your methods (Lines 140-142).

Second, they did not give any discussion on the section of trait diversity patterns (Line 256).

Third, what is the main question want to solve? To examine the relative importance of niche and neutral processes? If so, the authors said trait-abundance relationship was difficult to infer these processes? Then why the authors perform these analyses? I think to clarify the relationship between section 1 and section 2 is important.

Fourth, the Conclusion needs to be reframed and some conclusions of section 1 needs to be added.

Fifth, the authors used the rank of successional age as the treatment. However, the year is different for each successional age, how the authors explain these differences?

Minor comments:

Lines 68-71: the authors did not use the hydraulic conductivity traits.

Line 97: which one is easy-to-measure trait and which one is hard-to-measure trait?

Line 101: I think “the relative importance” may be more appropriate.

Line 88: add “is”.

Line 152: delete “Hence in this study”.

Lines 178-181: these might be redundant.

Tables 1 and 2: please re-organize the table (e.g., delete the line inside) and make it clear.

Line 191: only two traits significant in sites 2.

Lines 193-198: please re-write these results (e.g., which is negative and which is positive?)

Line 391: delete “-”,

Fig. 1: I think it is better to indicate the p values and significance in this figure, also to indicate the difference of these trait values.

Figs. 2 and 3: I think it is better to indicate which fitted lines in each panel were significant. What are the points in each panel?

Line 420: “The distributions of standard effect size (SES)” of what?

Reviewer #3: This paper addresses the several questions pertinent to community ecology about the relevance of species’ traits in determining species abundance. The authors have collected data on a number of traits related to carbon acquisition, stress tolerance, and reproduction on plants growing across two chronosequences following agricultural abandonment in alpine grasslands in China. A strength of the paper is the measurement of traits that are typically avoided due to their difficulty in assessing.

Overall the study was well-done. The manuscript was excellent in terms of experimental design and analysis. The study has specific goals and focused on an area which was not studied. The authors focused on interrelationships among successional age, functional traits, neutral-related traits, and species’ relative abundance. They also supported these observations by environmental characterization, intensive and prolonged UV radiation, the extremes of temperature, the short growing season in addition to the impact of these factors on some key physiological traits. These goals can help to find the differences between species relative abundance and the contribution of niche and neutral processes to species abundance during succession and abiotic stress tolerance. They collected the samples from high land 3000 m above sea level, they used different successional ages (4-, 6-, 10-, 13-years, and undisturbed for at least 40-years) and from a field. Their way of sample collection (two sites), random samples, size of the selected area and space also show data was measured in a proper way. Such things can convince us of the robustness of the statistical analysis.

The way of discussing the points was good; they discussed what their results mean in terms of ecology, stress tolerance and relationship between the species. Also, they express their opinion in all these issues and why they used the selected traits and how importantly these traits validate the hypothesis. They use most of the relevant references. Thus, I merely have some minor comments below:

Lines 32 Please change these sub-alpine meadow communities into these sub-alpine meadow communities during succession.

Lines 62-63 It is not so good to use shortcomings here. I do suggest to change to use three questions remains to be explored. Thus, please change the sentence “our current understanding of trait-abundance relationships suffers from at least two shortcomings that limit the predictable power of traits for community assembly processes” into “there are two following questions remains to be explored for current trait-abundance relationship studies”.

Line 96 Please change the sentence “Here we attempt to address the first shortcoming of trait-abundance studies by assembling data on six easy-to-measure morphological and hard-to-measure physiological traits (specific leaf area (SLA), seed mass, seed germination rate, height, leaf proline content and photosynthesis rate) to test trait-abundance relationships in a successional chronosequence of subalpine meadow plant” into “Here we attempt to utilize assembling data on six easy-to-measure morphological and hard-to-measure physiological traits (specific leaf area (SLA), seed mass, seed germination rate, height, leaf proline content and photosynthesis rate) to test trait-abundance relationships in a successional chronosequence of subalpine meadow plant communities”

Line 212 Please change “are support” into “support”

Line 235 Please change “is” into “was”

Line 237 Please change “appears” into “appeared”

Line 252 Please change “tend” into “tended”

Line 253 Please change “are” into “were”

Line 258 Please change “was” into “is”

Line 263 Please change “find” into “found”

Lines 267-275 Please check the tense of the Conclusion part, usually Conclusion should be written by past tense, as it has not published.

6. PLOS authors have the option to publish the peer review history of their article (what does this mean?). If published, this will include your full peer review and any attached files.

Reviewer #1: Yes: Zoltán Botta-Dukát

Reviewer #2: No

Reviewer #3: No

---

## [Author Response · Author response to Decision Letter 0]

28 Jan 2020

Comments from editor

1. In your Methods section, please provide additional information regarding the permits you obtained for the work. Please ensure you have included the full name of the authority that approved the field site access and, if no permits were required, a brief statement explaining why.

Response: As suggested, we have provided the additional information regarding which authority permits us to access the filed site in lines 144-146, pages 7-8.

Response: We do not make any changes to our Data Availability statement and will provide repository information for our data at acceptance. 

Reviewers’ comments: 

Reviewer #1: Understanding rules of community assembly is a central goal of community ecology, and trait-based models are useful tools to reach this goal. Therefore, the topic of this paper is interesting. Its further strength that the authors measured hard traits. Unfortunately, I have found so many problems in the manuscript that overwhelm the above mentioned advantages.

The logic of the Introduction and aims of the paper should be re-think. The coexistence of totally neutral species is stable, but not robust, because nearly neutral species cannot coexist. For more details on this topic see e.g. (Meszéna et al. 2006; Barabás et al. 2018). Thus, studying role of neutrality is meaningless, however some traits may prove to be neutral. Possibly, the authors mixed neutrality and stochasticity (it often occurs in the literature). Of course, exploring the relative role of stochastic and deterministic processes could be a valid aim. However, it would need different methodology, e.g. CATS developed by Shipley (Shipley et al. 2012; Shipley 2014). The applied methods allow quantifying importance of habitat filtering and limiting similarity, and its change during succession. Either the aims should be reformulated or different methods should be chosen.

Response: We agree that, nearly neutral species cannot exist in real communities, but neutral processes can still influence species abundance. Many studies have demonstrated that the mechanisms that inflfluence relative abundance are a subject of ongoing debate, currently reflflected in the dialogue on the importance of neutral versus niche-assembly processes in structuring communities (Hubbell 2001; Shipley, Vile & Garnier 2006; McGill et al. 2007; Levine & HilleRisLambers 2009; Cornwell and Ackely 2010). As per your recommendation, we have pointed out in lines 88-102, page 5, that our measured seed mass and seed germination rate are also highly related to neutral processes (dispersal and recruitment limitation). Thus we need to use SES to check whether FD is random (neutral processes) or not to further quantify whether neutral processes can affect variations in our measured functional traits. We agree that CATS developed by Shipley (Shipley et al. 2012; Shipley 2014) is a good method to disentangle the relative contributions of stochastic and deterministic processes to community assembly. However, many studies also demonstrated the SES can also help differentiate the relative role of stochastic and deterministic processes in community assembly over succession (Mason et al. 2012; Zhang et al. 2015; Pinho et al. 2018). Thus here we still use SES to quantify the relative contribution of stochastic and deterministic processes to community assembly along succession. 

Cornwell, W.K. & Ackerly, D.D. (2010) A link between plant traits and abundance: evidence from coastal California woody plants. Journal of Ecology, 98, 814-821. 

Hubbell SP. The unified neutral theory of biodiversity and biogeography. Princeton University Press. 2001.

Levine, J.M. & HilleRisLambers, J. (2009) The importance of niches for the maintenance of species diversity. Nature, 461, 254–257.

Mason NW, Richardson SJ, Peltzer DA, de Bello F, Wardle DA, Allen RB (2012) Changes in coexistence mechanisms along a long-term soil chronosequence revealed by functional trait diversity. Journal of Ecology 100, 678-689.

McGill, B.J., Etienne, R.S., Gray, J.S., Alonso, D., Anderson, M.J., Benecha, H.K. et al (2007) Species abundance distributions: moving beyond single prediction theories to integration within an ecological framework. Ecology Letters 10, 995-1015. 

Pinho, B. X., de Melo, F. P. L., Rodríguez, V.A., Pierce, S. Lohbeck, M. & Tabarelli, M. (2018). Soil-mediated filtering organizes tree assemblages in regenerating tropical forests. Journal of Ecology 106: 137-147.

Shipley, B., Vile, D. & Garnier, É. (2006) From plant traits to plant communities: a statistical mechanistic approach to biodiversity. Science, 314,812-814. 

Zhang H, Qi W, John R, Wang WB, Song FF, Zhou SR (2015) Using functional trait diversity to evaluate the contribution of multiple ecological processes to community assembly during succession. Ecography 38: 1147-1155.

I am missing the mentioning studies of community assembly along successional series from the Introduction. Moreover, in some references I think that the cited paper did not support well the statement (sometimes it did not say it at all, other cases it is not the main message of that paper and better citation could be found). For example, Ref3 in line 41; Ref8 in line 54; in line 55 paper that study changing trade-off along gradients also should be cited; Ref9 in line 61; Ref13 in line 74; Ref14 in line 78.

Response: As suggested, we have check the references citation problems and revised them accordingly. We hope there should be not any citation problems in our revised manuscript now. 

In the Methods, it remains unclear for me exactly how many plots were sampled and how they were arranged. It only mentioned later that two chronosequences were sampled.

Response: As per your suggestion, we have provided how many plots were sampled and how they were arranged in lines 131-142, page 7. 

I think the assumptions of ANCOVA (independence and normality of errors) were not satisfied. I suggest consulting with (Warton et al. 2015) and references therein. If abundance were measured by number of ramets, Poisson or negative binomial distribution would be suitable. Note, that in ANCOVA residuals should follow normal distribution, not the covariates.

Response: Our primary hypothesis is whether the relationship between traits and abundance shifts with age. The appropriate test for this is an analysis of covariance, with abundance as the dependent variable, age as a fixed, discrete factor and the trait as a continuous covariate. The test for a change in slope is based on the significance of the interaction term between age and the trait. our sampling of two independent chronosequences can further strengthen this analysis. We agree that ANCOVA residuals should follow normal distribution, not the covariates, thus we log-transformed both abundance and trait data to meet the normalization standard and we have provided this description in lines 169-172, page 9. Moreover, we performed the Shapiro-Wilk test and found that most the ANCOVA residuals could meet the normal distribution, after log-transforming both abundances and functional traits. 

To support your selection of randomization constrains please see (Götzenberger et al. 2016), and also see my paper (Botta-Dukát 2018) on possible problems of Gotelli & Grave’s SES.

Response: As suggested by your paper (Botta-Dukát 2018), we have firstly log-transformed both observed FD and FD random, then we calculate SES and we get the comparable FD patterns over succession (Fig. 4). We have also provided these responding descriptions in lines 200-203, page 10. 

Because correcting the above mentioned problems will lead to considerable changes in Results and Discussion, I stopped reviewing the manuscript at the end of Methods section.

Response: As suggested, we have tried our best to correct your mentioned problems above, and we hope currently revised version is clear now.

Cited references:

Barabás G, D’Andrea R, Stump SM (2018) Chesson’s coexistence theory. Ecol Monogr 88:277–303. doi: 10.1002/ecm.1302

Botta-Dukát Z (2018) Cautionary note on calculating standardized effect size (SES) in randomization test. Community Ecol 19:77–83. doi: 10.1556/168.2018.19.1.8

Götzenberger L, Botta-Dukát Z, Lepš J, et al (2016) Which randomizations detect convergence and divergence in trait-based community assembly? A test of commonly used null models. J Veg Sci 27:1275–1287. doi: 10.1111/jvs.12452

Meszéna G, Gyllenberg M, Pásztor L, Metz JAJ (2006) Competitive exclusion and limiting similarity: A unified theory. Theor Popul Biol 69:68–87. doi: 10.1016/j.tpb.2005.07.001

Shipley B (2014) Measuring and interpreting trait-based selection versus meta-community effects during local community assembly. J Veg Sci 25:55–65

Shipley B, Paine CET, Baraloto C (2012) Quantifying the importance of local niche-based and stochastic processes to tropical tree community assembly. Ecology 93:760–769. doi: 10.1890/11-0944.1

Warton DI, Shipley B, Hastie T (2015) CATS regression – a model‐based approach to studying trait‐based community assembly. Methods Ecol Evol 6:389–398. doi: 10.1111/2041-210X.12280

Reviewer #2: The authors examined the trait-abundance relationships and trait diversity patterns among successional stages in two meadow communities. The dataset of both traits and communities gives good opportunities to explore the ecological processes. However, there are some very important issues must be clarified.

Response: Thanks for your positive comments and pointed the weakness below.

First, for the analyses of trait-abundance relationship, the authors used the model: trait = abundance+age +abundance:age. I am very confused why abundance can predict the functional traits. Also, most parts of Introduction describe how traits can influence species abundance (I agree and this is reasonable). If the true model the authors wanted to use is abundance = trait + age + trait:age, then all relevant results needed to be revised. Also, the results are not consistent with your methods (Lines 140-142).

Response: As suggested, we have used your suggested ANCOVA model (abundance = trait + age + trait×age) to re-do the ANCOVA analysis. The new results please see Figs 3 and 4. We have also provided the detail descriptions of the ANCOVA model in lines 158-171, pages 8-9. 

Second, they did not give any discussion on the section of trait diversity patterns (Line 256).

Response: As suggested, we have provided the discussion on the FD patterns over succession in lines 296-301, page 15. 

Third, what is the main question want to solve? To examine the relative importance of niche and neutral processes? If so, the authors said trait-abundance relationship was difficult to infer these processes? Then why the authors perform these analyses? I think to clarify the relationship between section 1 and section 2 is important.

Response: As per your recommendation, we have re-framed our main questions in lines 110-119, page 6. Namely which life history strategies that are reflected by these 6 six traits can determine species abundance during succession; and whether FD is random or not along succession to check whether neutral processes can lead to variations in functional traits. We have also clarified the relationship between section 1 and section 2 in lines 88-102, page 5.

Namely “Although neutral processes do not predict any trait–abundance connection [6], any observed correlation between traits and abundance does not preclude the dominant influence of neutral processes on abundance either [5]. That is because, some traits (e.g., seed mass and seed germination rate) can also highly related to neutral processes (dispersal and recuritment limitation) [20, 21]. Functional trait diversity (FD) patterns are however more strongly determined by the nature of niche and neutral processes. So, if traits do not determine species presence/absence or abundance at a site, the diversity in trait values at the site should simply reflect that of a random sample from the larger species pool. On the other hand strong environmental filtering would select for a narrower range of trait values resulting in trait convergence. If resource competition were dominant, dissimilar species would be selected at a site causing trait divergence. Appropriate expectations can then be derived for FD patterns under purely neutral assembly (FDrandom), against which the observed FD patterns (FDobserved) can be compared. Thus testing FD patterns during succession can further facilitate to find out whether neutral or niche processes determine the variations in functional traits”. 

Fourth, the Conclusion needs to be reframed and some conclusions of section 1 needs to be added.

Response: As per your suggestion, we have provided conclusion of section 1 in lines 305-307, page 15. 

Fifth, the authors used the rank of successional age as the treatment. However, the year is different for each successional age, how the authors explain these differences?

Response: As suggested, we do not use rank of successional age, but treat successional age as a grouping factor that consisted of five levels. We have also pointed this out in lines 163-165, page 8.

Minor comments:

Lines 68-71: the authors did not use the hydraulic conductivity traits.

Response: As suggested, we have deleted this description.

Line 97: which one is easy-to-measure trait and which one is hard-to-measure trait?

Response: We have pointed out in lines 111-115, page 6, which one is easy-to-measure trait and which one is hard-to-measure trait.

Line 101: I think “the relative importance” may be more appropriate.

Response: We have rewritten this description.

Line 88: add “is”.

Response: Corrected as suggested in line 103, page 5.

Line 152: delete “Hence in this study”.

Response: As per your suggestion, we have deleted “Hence in this study”.

Lines 178-181: these might be redundant.

Response: As suggested, we have deleted this redundant content.

Tables 1 and 2: please re-organize the table (e.g., delete the line inside) and make it clear.

Response: Corrected as suggested in Tables 1 and 2.

Line 191: only two traits significant in sites 2.

Response: Corrected as suggested in line 216, page 11.

Lines 193-198: please re-write these results (e.g., which is negative and which is positive?)

Response: Corrected as suggested in lines 218-224, page 11.

Line 391: delete “-”,

Response: We have deleted “-”

Fig. 1: I think it is better to indicate the p values and significance in this figure, also to indicate the difference of these trait values.

Response: As suggested, we have provided p values in Fig. 1.

Figs. 2 and 3: I think it is better to indicate which fitted lines in each panel were significant. What are the points in each panel?

Response: As per your recommendation, we have pointed out in lines 483-485, page 28, each point represents the mean value of abundance and functional traits for a single species. However, ANCOVA model cannot provide the significance of each fitted, but it can demonstrated whether trait-abundance relationship vary significantly with successional age. 

Line 420: “The distributions of standard effect size (SES)” of what?

Response: Corrected as suggested in lines 495-496, page 28.

Reviewer #3: This paper addresses the several questions pertinent to community ecology about the relevance of species’ traits in determining species abundance. The authors have collected data on a number of traits related to carbon acquisition, stress tolerance, and reproduction on plants growing across two chronosequences following agricultural abandonment in alpine grasslands in China. A strength of the paper is the measurement of traits that are typically avoided due to their difficulty in assessing. Overall the study was well-done. The manuscript was excellent in terms of experimental design and analysis. The study has specific goals and focused on an area which was not studied. The authors focused on interrelationships among successional age, functional traits, neutral-related traits, and species’ relative abundance. They also supported these observations by environmental characterization, intensive and prolonged UV radiation, the extremes of temperature, the short growing season in addition to the impact of these factors on some key physiological traits. These goals can help to find the differences between species relative abundance and the contribution of niche and neutral processes to species abundance during succession and abiotic stress tolerance. They collected the samples from high land 3000 m above sea level, they used different successional ages (4-, 6-, 10-, 13-years, and undisturbed for at least 40-years) and from a field. Their way of sample collection (two sites), random samples, size of the selected area and space also show data was measured in a proper way. Such things can convince us of the robustness of the statistical analysis.The way of discussing the points was good; they discussed what their results mean in terms of ecology, stress tolerance and relationship between the species. Also, they express their opinion in all these issues and why they used the selected traits and how importantly these traits validate the hypothesis. They use most of the relevant references. Thus, I merely have some minor comments below:

Response: Thanks for your complement.

Lines 32 Please change these sub-alpine meadow communities into these sub-alpine meadow communities during succession.

Response: Corrected as suggested in line 34, page 2.

Lines 62-63 It is not so good to use shortcomings here. I do suggest to change to use three questions remains to be explored. Thus, please change the sentence “our current understanding of trait-abundance relationships suffers from at least two shortcomings that limit the predictable power of traits for community assembly processes” into “there are two following questions remains to be explored for current trait-abundance relationship studies”.

Response: Corrected as suggested in lines 71-73, page 4.

Line 96 Please change the sentence “Here we attempt to address the first shortcoming of trait-abundance studies by assembling data on six easy-to-measure morphological and hard-to-measure physiological traits (specific leaf area (SLA), seed mass, seed germination rate, height, leaf proline content and photosynthesis rate) to test trait-abundance relationships in a successional chronosequence of subalpine meadow plant” into “Here we attempt to utilize assembling data on six easy-to-measure morphological and hard-to-measure physiological traits (specific leaf area (SLA), seed mass, seed germination rate, height, leaf proline content and photosynthesis rate) to test trait-abundance relationships in a successional chronosequence of subalpine meadow plant communities”

Response: Corrected as suggested in lines 112-116, page 6.

Line 212 Please change “are support” into “support”

Response: Corrected as suggested in line 238, page 12.

Line 235 Please change “is” into “was”

Response: Corrected as suggested in line 264, page 13.

Line 237 Please change “appears” into “appeared”

Response: Corrected as suggested in line 267, page 13.

Line 252 Please change “tend” into “tended”

Response: Corrected as suggested in line 283, page 14.

Line 253 Please change “are” into “were”

Response: Corrected as suggested in line 284, page 14.

Line 258 Please change “was” into “is”

Response: Corrected as suggested in line 291, page 14.

Line 263 Please change “find” into “found”

Response: We have rewritten this sentence.

Lines 267-275 Please check the tense of the Conclusion part, usually Conclusion should be written by past tense, as it has not published.

Response: Corrected as suggested in lines 303-311, page 15.

---

## [Decision Letter · Decision Letter 1]

7 Feb 2020

Niche-processes induced differences in plant growth, carbon balance, stress resistance, and regeneration affect community assembly over succession

PONE-D-19-21921R1

Dear Dr. Zhang,

We are pleased to inform you that your manuscript has been judged scientifically suitable for publication and will be formally accepted for publication once it complies with all outstanding technical requirements.

With kind regards,

Craig Eliot Coleman, PhD

Academic Editor

PLOS ONE

Additional Editor Comments (optional):  One of your reviewers suggested some additional corrections which you should consider as you prepare the final manuscript for publication but I did not feel they were enough to warrant having to revise and resubmit.

Reviewers' comments:

Reviewer's Responses to Questions

**Comments to the Author**

1. If the authors have adequately addressed your comments raised in a previous round of review and you feel that this manuscript is now acceptable for publication, you may indicate that here to bypass the “Comments to the Author” section, enter your conflict of interest statement in the “Confidential to Editor” section, and submit your "Accept" recommendation.

Reviewer #2: All comments have been addressed

Reviewer #3: All comments have been addressed

2. Is the manuscript technically sound, and do the data support the conclusions?

Reviewer #2: Yes

Reviewer #3: Yes

3. Has the statistical analysis been performed appropriately and rigorously? 

Reviewer #2: Yes

Reviewer #3: Yes

4. Have the authors made all data underlying the findings in their manuscript fully available?

Reviewer #2: Yes

Reviewer #3: Yes

5. Is the manuscript presented in an intelligible fashion and written in standard English?

Reviewer #2: Yes

Reviewer #3: Yes

6. Review Comments to the Author

Reviewer #2: Thanks for the careful revisions of the authors. I think there is a great improvement of this manuscript. There I have only few minor comments:

Line 165: please add an error term

Tables 1-2: I think it is better to give the information of coefficients instead of F value

Line 481: I will say each point is one species.

Figs. 2 and other: I suggest only adding the significant lines.

Fig. 1. Please added the statistical methods or in the fig legend text.

Fig. 4. Although the authors give the SES values of all traits-based FD, there are much information in individual-trait based SES.FD patterns.

Reviewer #3: The authors have done a fine job of responding to all of my comments. I congratulate the Authors for producing this paper and I look forward to reading it in print.

7. PLOS authors have the option to publish the peer review history of their article (what does this mean?). If published, this will include your full peer review and any attached files.

Reviewer #2: No

Reviewer #3: No

---

## [Editor Report · Acceptance letter]

14 Feb 2020

PONE-D-19-21921R1 

Niche-processes induced differences in plant growth, carbon balance, stress resistance, and regeneration affect community assembly over succession 

Dear Dr. Zhang:

I am pleased to inform you that your manuscript has been deemed suitable for publication in PLOS ONE. Congratulations! Your manuscript is now with our production department. 

With kind regards,

on behalf of

Dr. Craig Eliot Coleman 

Academic Editor

PLOS ONE